# Limb Salvage after Lower-Leg Fracture and Popliteal Artery Transection—The Role of Vessel-First Strategy and Bone Fixation Using the Ilizarov External Fixator Device: A Case Report

**DOI:** 10.3390/medicina57111220

**Published:** 2021-11-09

**Authors:** Vincenzo Giordano, Felipe Serrão Souza, William Dias Belangero, Robinson Esteves Pires

**Affiliations:** 1Serviço de Ortopedia e Traumatologia Prof. Nova Monteiro, Hospital Municipal Miguel Couto, Rio de Janeiro 22430-060, Brazil; felipeserrao@yahoo.com.br; 2Departamento de Ortopedia, Universidade Estadual de Campinas, Campinas 13083-887, Brazil; belangerowd@gmail.com; 3Departamento de Ortopedia, Universidade Federal de Minas Gerais, Belo Horizonte 31270-901, Brazil; robinsonestevespires@gmail.com

**Keywords:** popliteal artery, limb ischemia, external fixation, Ilizarov circular fixator, malunion

## Abstract

Open traumatic lesion of the popliteal artery is relatively rare. Ischemia time longer than 6 h and severity of limb ischemia have been shown to be associated with an increased risk of limb loss. Severe local infection is critical in the presence of major soft tissue trauma or open fractures. We report the case of a young female who suffered a traumatic transection of the popliteal artery associated with an open fracture of the distal tibia and fibula managed by direct vessel reconstruction with an end-to-end repair and skeletal stabilization initially with half-pin external fixation, then replaced by an Ilizarov circular frame. The patient had a very satisfactory outcome, but the fracture healed malunited, later corrected by open reduction and internal fixation with lag-screwing and a neutralization plate.

## 1. Introduction

Open traumatic lesion of the popliteal artery is relatively rare, being associated with a high risk of limb loss, with an amputation rate ranging from 10% to 16% [1,2]. Ischemia time longer than 6 h and severity of limb ischemia have been shown to be associated with an increased risk of limb loss [1,3]. Furthermore, severe local infection, which is critical in the presence of major soft tissue trauma or open fractures, has been shown to jeopardize the outcome [1,4]. Rapid diagnosis and prompt surgery decrease the period of ischemia and the rate of amputation; thus, a multidisciplinary team approach has been suggested as a key factor for limb salvage [1,4]. Revascularization, either including temporary vascular shunting or not, and protection, preferably using external fixation devices for bone stabilization, have been recommended in these cases [1]. In this scenario, the main benefit of external fixation devices is the ability to stabilize the fracture in a minimally traumatic way, further reducing soft tissue damage and potentially protecting the vascular repair [5,6,7]. Herein, we describe the case of a young female, who suffered an associated traumatic popliteal artery transection and open tibia fracture treated with a vessel-first approach and Ilizarov circular external fixation, with successful revascularization of the leg and adequate bone healing.

## 2. Case Report

An 18-year-old female was admitted to the Emergency Department (ED) after being hit by a bus in front of our hospital. On arrival, her vital signs were as follows: blood pressure 110/70 mmHg, pulse rate 91/min, respiratory rate 18/min, and body temperature 36.5 °C. Initial laboratory exams were as follows: hematocrit—33%, hemoglobin—10.0 g/dL, and lactate–−1.5, which were interpreted as characteristic of moderate acute bleeding (hemorrhagic shock class 2). After the initial evaluation in the ED, she was noted to have a cold, pale, pulseless left foot with absent popliteal and ankle pulses. She had a complete degloving of the entire posterior part of the left leg, with exposure of the gastrocnemius muscle from the popliteal region to the distal third of this leg. No other skeletal or non-skeletal injuries were noted on the initial examination. Initial radiographs of the left leg and knee revealed a transverse fracture of the distal tibia and fibula, suggesting a rolling mechanism of the bus wheel as it passed over her leg. She was diagnosed as having a traumatic transection of the popliteal artery associated with an open distal tibia shaft and fibula fracture. A tourniquet was applied to her left thigh to restrict blood flow to the injured limb, thus potentially preventing life-threatening blood loss, and she was immediately taken to the Operating Theatre (OT), where she underwent general anesthesia with an orotracheal tube. The decision was taken as a protective ventilation strategy. In the OT, she had a secondary examination of the left leg, and the injuries were confirmed by direct inspection (Figure 1). Due to the short ischemia time (less than 1 h), the tourniquet was kept in place, and the initial debridement was carried out with the patient in the supine position. Then, the distal tibia and fibula fracture was reduced, and a half-pin spanning external fixator was applied on the anterior aspect of the left lower limb, with the knee in slight flexion (approximately 10°) and the ankle in neutral position. The initial surgical management took no longer than 30 min. The patient was re-positioned in the prone position for the second part of the procedure. Vascular surgeons performed a thorough exploration of the popliteal vessels, tibial nerve, and their branches, noting a complete transection in the middle part of the popliteal artery, which was repaired by direct vessel reconstruction with an end-to-end repair. Due to the slight flexion of the knee, a tension-free anastomosis was possible, with no need for interposition graft. The open wound was thoroughly irrigated, and the skin flaps were loosely approximated and sutured to the gastrocnemius muscle (Figure 2). A deep posterior compartment fasciotomy was performed, but no further decompression was required for the anterior and lateral compartments of the leg. A bulky Jones dressing was applied, as the negative pressure wound therapy (NPWT) was not available in the hospital. There was no need for inotropes or vasopressors during surgery. At the end of the procedure, her vital signs were as follows: blood pressure 115/80 mmHg, pulse rate 69/min, respiratory rate 14/min, and body temperature 36.5 °C. The patient was sent to the Intensive Care Unit (ICU) for sequential care.

On day 2, leg and pedal pulses were present as noted on clinical examination and arterial Doppler study. The patient was taken to the OT for a second look, with a large area of necrosis noted in the antero-medial part of the left leg. The skin flap used to cover the degloved area of the leg was completely removed, and the wound was extensively debrided and irrigated (Figure 3). A bulky Jones dressing was applied, and she was taken to the ICU. As the vascular injury progressed satisfactorily, there was no longer any need for intervention by the vascular surgery team. The patient was transferred to the Orthopedic ward on day 7. She had sequential debridement every 48 to 72 h until the wound was considered adequate and there was no further risk of amputation either due to ischemia or infection. By day 21, the half-pin external fixator was exchanged by another one, not anymore spanning the knee joint (Figure 4). On day 30, the half-pin external fixator was removed, and an Ilizarov circular frame was applied to hold the distal tibia in place until bone healing. No wound closure either using NPWT, flaps, or skin graft was done (Figure 5). The patient remained hospitalized for continuous care and initiation of the rehabilitation protocol, including flat-foot weight bearing with crutches. There were no signs of ischemia or vascular insufficiency on the injured lower limb.

On day 60, the patient was discharged from the hospital with the Ilizarov fixator, showing obvious signs of adequate soft tissue healing of the extensive leg wound. Radiographs showed progression of bone healing, with a slight varus on the fracture site (Figure 6). After 100 days since the index surgery, she had no distal neurovascular deficit, the tibia fracture and the leg wound were totally healed, and the patient was admitted for removal of the Ilizarov frame (Figure 7). Radiographs showed malunited fractures of the left distal tibia and fibula, with 12° varus, 8° antecurvatum, and 5 mm shortening, which were judged unacceptable. The decision was for a scheduled corrective osteotomy without Ilizarov to fulfil the patient’s request.

The patient was re-admitted after 30 days for a one-stage corrective osteotomy of the malunited fractures. A 2 cm resection of the distal fibula was initially performed through a lateral longitudinal incision, followed by an oblique osteotomy of the distal tibia through an antero-medial approach. After anatomic alignment of the tibia axis in both planes, the osteotomy was fixed with a 3.5 mm cortical lag-screw and a neutralization small fragment non-locked reconstruction plate (Figure 8). The osteotomy site healed uneventfully and with perfect alignment after 3 months, with no wound problem. At the last follow-up at 5 years, the range of motion of her left ankle and knee were completely recovered, except for a slight decrease (< 10°) in both ankle dorsiflexion and subtalar movement. There were no signs of arthritis. She complained of slight pain after extreme use, with normal function in work-related activities, and a normal gait without limping. Last consultation radiographs revealed a satisfactory alignment of the left leg on both planes (Figure 9). According to the Reidsma et al. [8] outcome score, she was rated with a good subjective and objective result: implicit mild pain with overuse, normal function in work-related activities, but restriction of strenuous activities.

## 3. Discussion

We report the case of an 18-year-old female patient who suffered an extensive leg degloving with a distal tibia and fibula fracture and a popliteal artery transection. This is a typical and educational case that demonstrates the importance of the rapid diagnosis and multidisciplinary team approach to decrease the period of ischemia and the rate of amputation [1,4]. A limb ischemia greater than 6 h results in severe necrosis and amputation in up to 30% of patients [3,9]. Additionally, compartment syndrome is a major risk factor for amputation following artery injury; thus, fasciotomy done at the time of arterial repair lowers amputation rates [10,11]. Despite the extensive degloving observed in our patient, the deep posterior compartment of her left leg was decompressed, as recommended. One of the main points of debate is related to the sequence of approach to vascular and skeletal injuries. Due to the straight limit of ischemia time, revascularization either with direct vessel reconstruction or intra-vascular shunt must be performed prior to skeletal stabilization whenever possible [1]. Nevertheless, the multidisciplinary team must be aware that, in selected patients with severely bleeding fractures, orthopedic stabilization prior to vascular repair present benefits without increased risk of limb loss [1,11]. Due to the short ischemia time, our patient was managed with initial debridement and skeletal stabilization using a half-pin spanning external fixator first.

External fixation devices have been shown to be less invasive, achieve adequate stability, and provide good access for wound management without compromising stability [11]. In addition, a controlled environment protects the vascular repair from disruption [12,13]. In our patient, we preferred to initially use an external half-pin device due to its safety, versatility, and quick application. A half-pin external fixator was maintained for skeletal stabilization until vascular surgeons defined that the risk of revascularization-related complications was minimal; thus, other orthopedic interventions could be performed, and the half-pin external fixator was replaced by the Ilizarov frame. The decision to apply the circular external fixator was to hold the distal tibia in place until bone healing. The option for a more stable construction permitted the initiation of a more vigorous rehabilitation protocol, including weight bearing, which has been shown to have a direct relationship with the improvement of severe soft tissue injuries, reducing the inflammation and allowing joint mobilization [14].

During follow-up, the patient had no neurovascular deficit or signs of ischemia on the left leg, and the distal tibia and fibula fracture uneventfully healed with some angulation and shortening after 100 days. The surgical correction was done using an oblique single-cut osteotomy fixed with a 3.5 mm lag-screw and a neutralization plate. In the last follow-up, the patient was rated with a good subjective and objective outcome, mainly due to slight pain with excessive use and certain restriction of strenuous activities. It is well known that fractures of the shaft of the tibia often heal with some angulation, and although the amount of residual deformity that can be accepted is controversial, varus malalignment has been recognized as a cause of osteoarthritis in the medial compartment of the knee [15,16]. Some authors reported a rate of malunion with Ilizarov fixator ranging from 2% to 21.5% [17,18,19], probably due to either inadequate reduction in the fracture before fixation or loss of wire tensioning and subsequent loss of reduction [16,20]. Li et al. [21] performed a systematic review and meta-analysis of peer-reviewed published studies on tibial plateau fractures treated with either open reduction with internal fixation (ORIF) or circular external fixation to compare functional, radiological outcomes, postoperative complications, and reoperation rates between the two methods. Five of 17 studies reported data on malunions. Their findings indicated that circular external fixator compared to ORIF resulted in a significantly higher chance of developing malunion (OR, 2.56; 95% CI, 1.12–5.84; I2 = 49%; *p* = 0.03). Whether this is attributable to the treatment itself, the thin-wire circular device, or the severity of the injury was unclear, a prolonged healing time for tibial fractures treated with external fixators has been reported in the literature, which may lead to an increased risk of loss of wire pre-tension, potentially affecting the process of fracture healing [22,23]. It has been shown that changes in the mechanical properties of a fracture site affect the time taken for a fracture to heal, thus modifying the type and proportion of tissues formed [24]. In addition, poor new bone formation and premature removal can lead to axial deformity and re-fracture of the new bone [25,26]. Nevertheless, although the Ilizarov external fixator is an established technique for both achieving bone union and treating bone-healing disturbances, it is extremely important to recognize some complications that can occur with this method.

## 4. Conclusions

In summary, we report the case of a young female that suffered a traumatic transection of the popliteal artery associated with an open fracture of the distal tibia and fibula managed by direct vessel reconstruction with an end-to-end repair and skeletal stabilization initially with half-pin external fixation, then replaced by an Ilizarov circular frame. The patient had a satisfactory outcome according to the Reidsma score, but the fracture healed in an unacceptable malalignment, later corrected by open reduction and internal fixation with lag-screwing and a neutralization plate. The learning points of this case report are as follows: (1) it is critical to rapidly revascularize the segment after a complete injury of the popliteal artery; (2) direct vessel reconstruction leads satisfactory outcomes when the repair is done as quick as possible; (3) the use of external fixation is effective, fast, versatile, and safe for the initial management of severe open injuries of the lower extremities; (4) the Ilizarov circular frame is adequate for holding the bone in place until union, but there is a risk for nonunion in up to 25% of the cases; (5) the amount of residual deformity that can be accepted after tibia fractures is controversial, though varus malalignment is a potential cause of osteoarthritis in the medial compartment of the knee and requires corrective osteotomy.

## Figures and Tables

**Figure 1 medicina-57-01220-f001:**
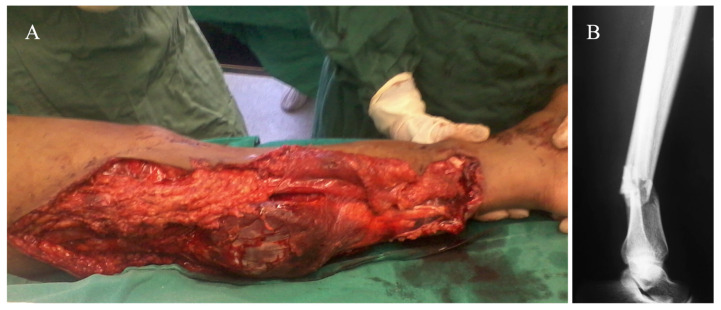
(**A**) Severe degloving of the left leg. (**B**) Lateral radiograph of the left leg, showing an open simple transverse distal tibia and fibula fracture.

**Figure 2 medicina-57-01220-f002:**
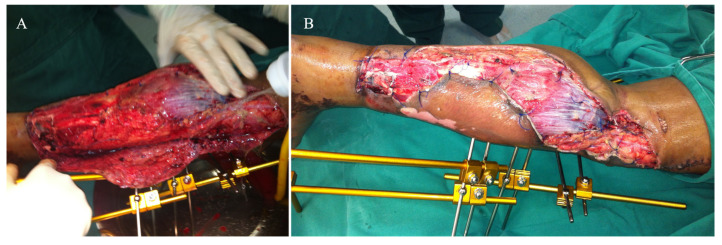
(**A**) The open wound was initially debrided, and a half-pin spanning external fixator was applied on the anterior aspect of the left lower limb, with the knee in slight flexion (approximately 10°) and the ankle in neutral position. (**B**) After identification of the complete transection in the middle part of the popliteal artery, vascular surgeons performed direct vessel reconstruction with an end-to-end repair, and the skin flaps were loosely approximated and sutured to the gastrocnemius muscle.

**Figure 3 medicina-57-01220-f003:**
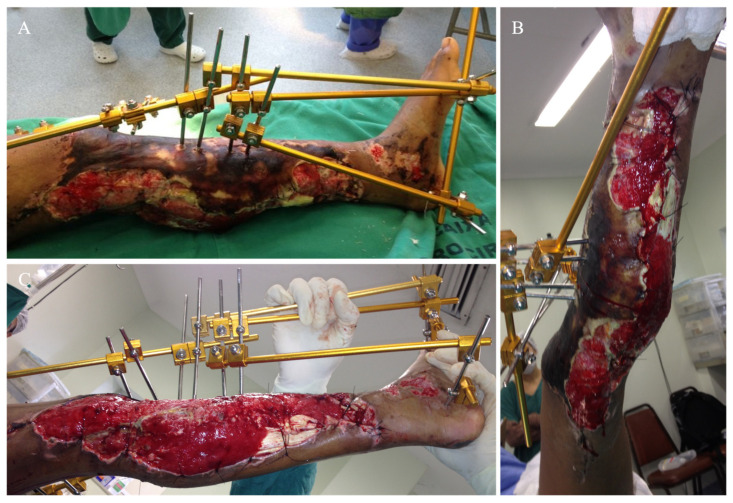
(**A**,**B**) Anterior and medial aspects of the left leg on day 2, demonstrating a large area of necrosis in the antero-medial part of the left leg. (**C**) The skin flap used to cover the degloved area of the leg was completely removed, and the wound was extensively debrided and irrigated.

**Figure 4 medicina-57-01220-f004:**
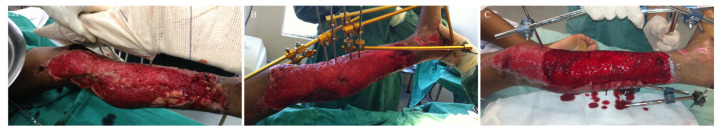
(**A**) Aspect of the leg on day 4. (**B**) Aspect of the leg on day 12. (**C**) Aspect of the leg on day 30, when half-pin external fixator was removed and an Ilizarov circular frame was applied to hold the distal tibia in place until bone healing.

**Figure 5 medicina-57-01220-f005:**
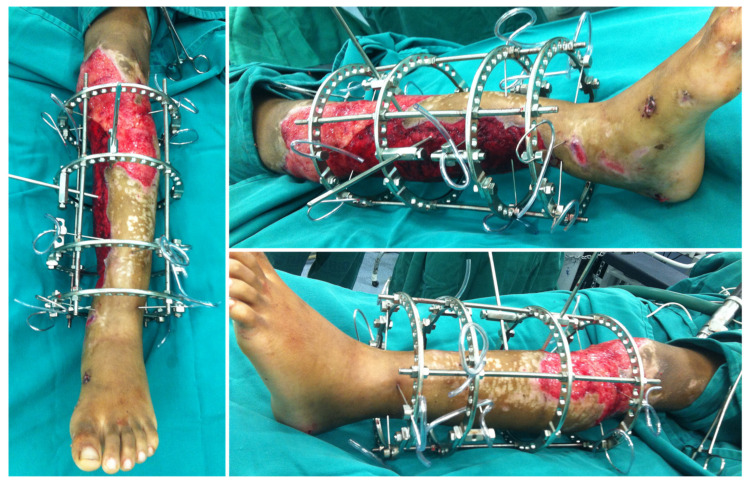
Aspect of the leg one week after the Ilizarov circular frame was applied. No wound closure either using NPWT, flaps, or skin graft was done. There were no signs of ischemia or vascular insufficiency.

**Figure 6 medicina-57-01220-f006:**
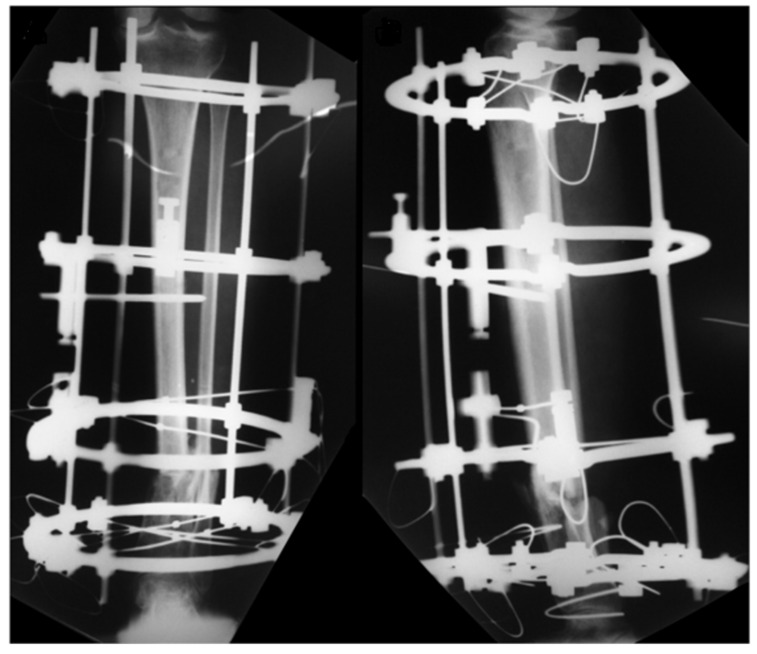
AP and lateral radiographic views of the left leg demonstrating the adequate alignment of the fracture and slight progression of bone healing, with minimum varus on the fracture site.

**Figure 7 medicina-57-01220-f007:**
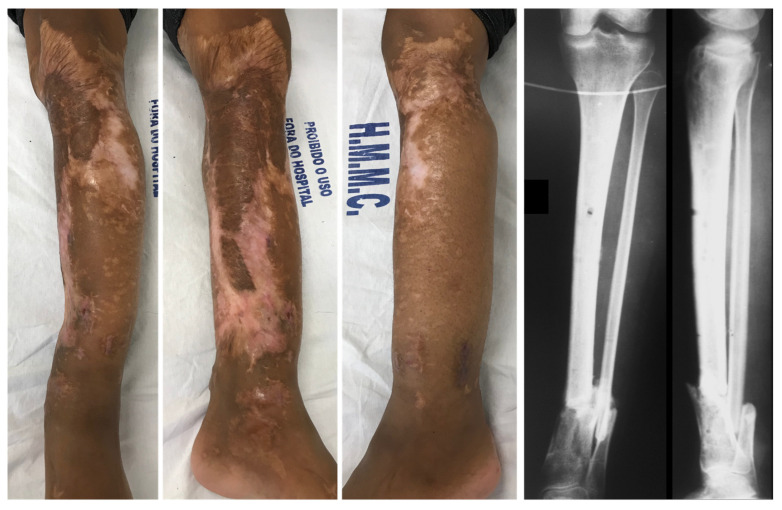
On day 100, the fracture and the leg wound were totally healed, and the patient was admitted for removal of the Ilizarov frame. Radiographs showed malunited fractures of the left distal tibia and fibula.

**Figure 8 medicina-57-01220-f008:**
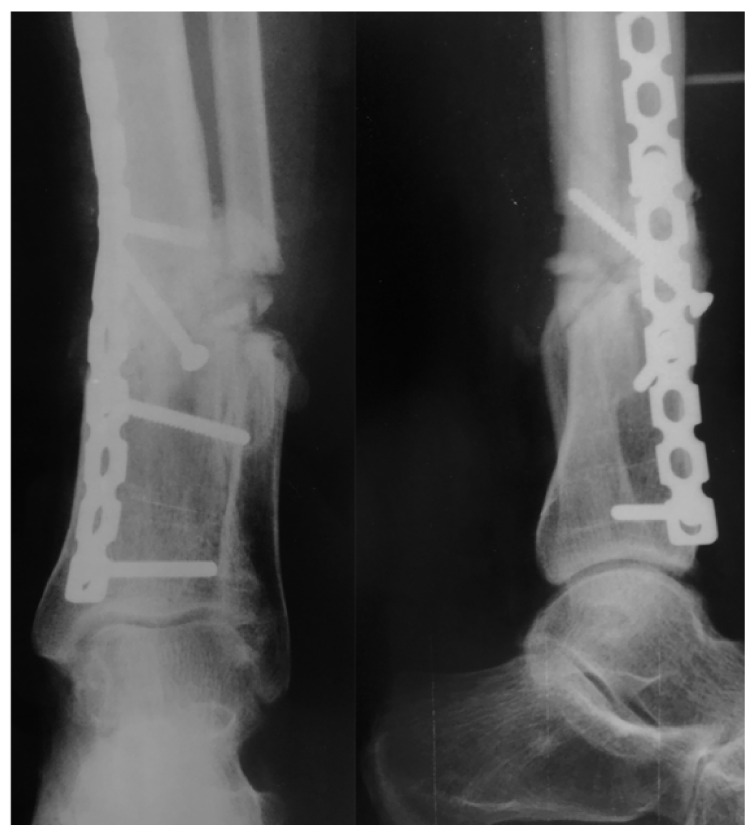
Immediate postoperative radiographic images of the one-stage corrective osteotomy of the malunited fractures. A 2 cm resection of the distal fibula was initially performed, followed by an oblique osteotomy of the distal tibia through an antero-medial approach.

**Figure 9 medicina-57-01220-f009:**
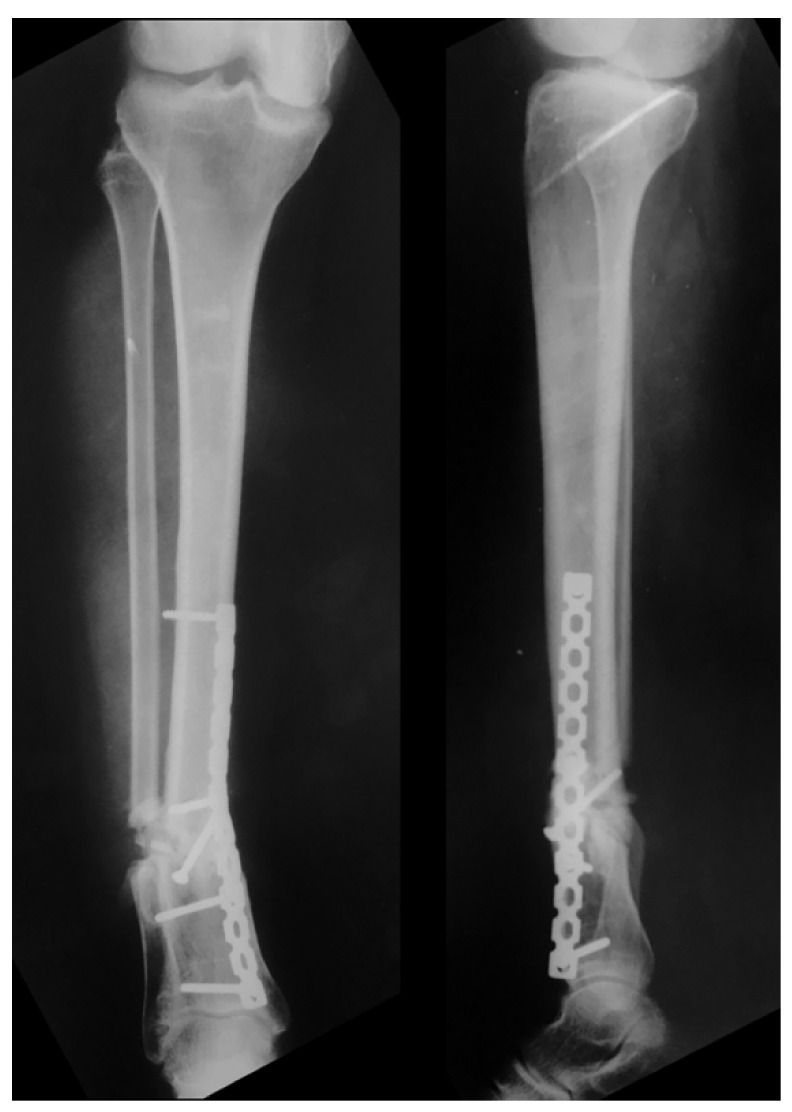
Radiographs taken in the last follow-up revealed a satisfactory alignment of the left leg on both planes.

## Data Availability

Data and materials are available on reasonable request.

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
