# Peer review of "Limb Salvage after Lower-Leg Fracture and Popliteal Artery Transection—The Role of Vessel-First Strategy and Bone Fixation Using the Ilizarov External Fixator Device: A Case Report"

_medicina, 2021, doi:10.3390/medicina57111220_

Round 1

Reviewer 1 Report

  • Would appreciate pictures of popliteal artery before/after reconstruction.
  • It is not clear.. the patient was actively bleeding from the artery, thus the need for the tourniquet?
  • If the patient was actively bleeding, than her life was immediately threatened, not just limb viability. This should be clarified. Hb at admission and after surgery is of interest. Did the patient require blood transfusions?
  • Please add type of anesthesia chosen for the patient and why.
  • Some healing devices were not available, but what about skin/ wound management from a plastic surgeon's point of view, at a very young female patient?

Author Response

Response to Reviewer 1 Comments

Point 1: Would appreciate pictures of popliteal artery before/after reconstruction.

Response 1: Thanks for your comment. Unfortunately, due to the acute scenario and because this stage of the treatment was performed by the vascular surgery team, there is no photographic documentation of the lesion, only description in the patient’s medical chart

Point 2: It is not clear.. the patient was actively bleeding from the artery, thus the need for the tourniquet?

Response 2: Again, thanks for you input. Yes, it was necessary the tourniquet to restrict blood flow to the injured limb, thus potentially preventing life-threatening blood loss. This information was added to the main text.

Point 3: If the patient was actively bleeding, than her life was immediately threatened, not just limb viability. This should be clarified. Hb at admission and after surgery is of interest. Did the patient require blood transfusions?

Response 3: Patient present an acute severe bleeding; therefore, the tourniquet was applied to restrict blood flow to the injured limb in the Emergency Room during initial evaluation and resuscitative manoeuvres. On arrival, she presented a blood pressure of 110/70 mmHg, pulse rate of 91/min, respiratory rate of 18/min, and body temperature of 36.5°C, not characterizing a massive bleeding. Initial laboratory exams were as follows: Haematocrit – 33%, Hemoglobin – 10.0 g/dL, and Lactate – -1.5, which were interpreted as characteristic of moderate acute bleeding (haemorrhagic shock class 2), probably rapidly compensated in a young patient with adequate physiologic reserve. This information was added to the main text.

Point 4: Please add type of anesthesia chosen for the patient and why.

Response 4: Patient underwent general anaesthesia with an orotracheal tube. The decision was taken as a protective ventilation strategy. This information was added to the main text.

Point 5: Some healing devices were not available, but what about skin/ wound management from a plastic surgeon's point of view, at a very young female patient?

Response 5: Thanks for your comment. This certainly could be an option. However, we decided to keep all care with the Orthopaedic team, in addition to the Vascular Surgery team. Nevertheless, this is a very important aspect of the multidisciplinary approach, which is nowadays more widely recognized as the Orthoplastic approach.

Reviewer 2 Report

This is an interesting case report

however some grammatical errors persist, which should be rectified

Author Response

Response to Reviewer 2 Comments

Point 1: This is an interesting case report, however some grammatical errors persist, which should be rectified.

Response 1: Thanks for your comment. We reviewed the entire text to correct all grammatical errors we could observe. As no author is primarily English-native, we used an English revisor. We hope the grammar improved.
